# Parylene C as a Multipurpose Material for Electronics and Microfluidics

**DOI:** 10.3390/polym15102277

**Published:** 2023-05-12

**Authors:** Beatriz J. Coelho, Joana V. Pinto, Jorge Martins, Ana Rovisco, Pedro Barquinha, Elvira Fortunato, Pedro V. Baptista, Rodrigo Martins, Rui Igreja

**Affiliations:** 1CENIMAT|i3N, Department of Materials Science, NOVA School of Science and Technology, Campus de Caparica, NOVA University of Lisbon and CEMOP/UNINOVA, 2829-516 Caparica, Portugal; bj.coelho@campus.fct.unl.pt (B.J.C.);; 2UCIBIO, I4HB, Department of Life Sciences, NOVA School of Science and Technology, Campus de Caparica, NOVA University of Lisbon, 2829-516 Caparica, Portugal

**Keywords:** Parylene C, XRD characterization, thermal characterization, electronic devices, dielectric, encapsulation, substrate

## Abstract

Poly(p-xylylene) derivatives, widely known as Parylenes, have been considerably adopted by the scientific community for several applications, ranging from simple passive coatings to active device components. Here, we explore the thermal, structural, and electrical properties of Parylene C, and further present a variety of electronic devices featuring this polymer: transistors, capacitors, and digital microfluidic (DMF) devices. We evaluate transistors produced with Parylene C as a dielectric, substrate, and encapsulation layer, either semitransparent or fully transparent. Such transistors exhibit steep transfer curves and subthreshold slopes of 0.26 V/dec, negligible gate leak currents, and fair mobilities. Furthermore, we characterize MIM (metal–insulator–metal) structures with Parylene C as a dielectric and demonstrate the functionality of the polymer deposited in single and double layers under temperature and AC signal stimuli, mimicking the DMF stimuli. Applying temperature generally leads to a decrease in the capacitance of the dielectric layer, whereas applying an AC signal leads to an increase in said capacitance for double-layered Parylene C only. By applying the two stimuli, the capacitance seems to suffer from a balanced influence of both the separated stimuli. Lastly, we demonstrate that DMF devices with double-layered Parylene C allow for faster droplet motion and enable long nucleic acid amplification reactions.

## 1. Introduction

Poly(chloro-para-xylylene), or Parylene C, is a flexible dielectric polymer belonging to the poly(p-xylylene) family [1,2]. This polymer is widely used due to its unique set of properties, such as chemical inertness, transparency, flexibility, conformability (also due to the deposition process) [3,4], and dielectric properties. Commonly employed either as a substrate or for encapsulation, Parylene C provides a flexible medium in which electronic signals can operate up to very elevated frequencies due to its low loss properties [5,6], while its low moisture absorption [7] substantially enhances the stability of compatible electronic technologies, such as oxides and organics. Parylene C can also be used as a dielectric layer in electronic devices, such as capacitors [8,9,10] or transistors [11,12,13]. While having a lower dielectric permittivity (*𝜀_r_* = 3.2) than other, more conventional dielectrics, its flexibility, good dielectric strength, and insulation properties can provide high device reliability when considering applications under strain [14]. Using Parylene C for electronic applications enables the production of organic, fully transparent, and flexible devices [2]. Such devices are particularly relevant for new applications, such as display technology, wearables, or even skin electronics and implantable electronics for human-device interfaces [15,16,17]. In fact, Parylene C has been approved by the Food and Drug Administration (FDA) as a US Pharmacopeial Convention (USP) class IV biocompatible polymer [7,18], and has ever since been applied to a plethora of fields in life sciences, such as prostheses [19,20], neural implants [18,21], and cell growth [22].

Specifically regarding the field of thin-film transistors (TFTs), Parylene C has attracted the attention of the scientific community, considering its multifunctional applications within the structure of TFTs: Parylene C can be used as a substrate [11,23,24], as a dielectric [12,25,26], and as a passivation layer [27,28,29]. Using Parylene C as a TFT substrate is highly important to produce flexible devices, thus taking one step forward toward the flexibilization of electronics. Moreover, as previously stated, one appealing property of Parylene C is its transparency; therefore, TFTs fabricated on Parylene C substrates will also benefit from being optically transparent. As a dielectric layer, Parylene C is also a promising candidate, providing TFTs with a uniform and tendentially pinhole-free dielectric layer, with excellent resistance to breakdown (and therefore, excellent dielectric strength) [7,26]. The dielectric layer on TFTs also benefits from the high purity of the deposited layers due to the unique chemical vapor deposition (CVD) process used for Parylene C coatings. This process enables the production of Parylene C directly from its dimer form, and does not require any additional material, either as a catalyst or for pre- or post-processing, hence the high degree of purity [30]. Finally, Parylene C is a good encapsulation material due to its low gas and moisture permeability, as well as its conformal deposition. Herein, we present TFTs featuring trivalent Parylene C as a substrate, dielectric, and encapsulation layer and further compare device characteristics in semitransparent and fully transparent configurations (with metallic and transparent oxide electrodes, respectively).

Another interesting application for this polymer is digital microfluidics, where Parylene C is commonly used as a dielectric [31]. Briefly, digital microfluidics enable the motion of low-volume droplets over an electrode array. Several methodologies enable said motion (e.g., surface acoustic waves [32], optoelectrowetting [33], and magnetic actuation [34]), but for one of the most common approaches—electrowetting—a dielectric material is strictly necessary to prevent water electrolysis due to the use of a strong electric field [35]. With this in mind, our group invested in Parylene C as the standard dielectric material for digital microfluidic (DMF) platforms, which are specifically designed to perform isothermal nucleic acid amplification (NAA) [36,37,38]. Herein, we explore the influence of two relevant aspects of DMF devices for NAA: temperature and actuation signal. Considering that the breakdown of the dielectric is a major constraint in DMF [39], we further investigate the breakdown prevention strategy used by our group, which consists of depositing Parylene C in two independent layers, as opposed to a single, thicker layer. Resorting to this strategy, we aim to reduce the probability of dielectric breakdown. To characterize the response of the dielectric layer to temperature and voltage signal stimuli, we created MIM (metal–insulator–metal) structures based on Parylene C (single- and double-layered), thus producing a simplified device for testing the dielectric layer. Lastly, we demonstrate that DMF devices with double-layered Parylene C are adequate for long NAA reactions and enable faster droplet motion.

## 2. Materials and Methods

All Parylene C thin films were deposited by CVD using the Labcoater^®^-PDS 2010 equipment (Specialty Coating Systems, Indianapolis, IN, USA), from Parylene C precursor dimer (Specialty Coating Systems, Indianapolis, IN, USA; CAS 28804-46-8). Low adhesion to substrates is a known issue for Parylene C; therefore, adhesion promoters are often used. Herein, unless stated otherwise, three to four drops of silane adhesion promoter (Silane A 174—Merck, Darmstadt, Germany) were added to the deposition chamber immediately before the deposition itself. Silane promotes adhesion by creating covalent bonds between Parylene C radicals and glass or other materials’ surfaces [40,41]. Different Parylene C layers were produced by changing the amount of Parylene C dimer in the evaporation chamber. Dimer masses between 0.2 g and 30 g were used throughout this work, with thicknesses ranging from nearly 200 nm to 20 μm. For all the produced devices, unless stated otherwise, Parylene C was deposited in a single deposition process, i.e., as a single-layer structure.

### 2.1. Parylene C Structural, Thermal, and Electrical Characterization

The structural characterization of Parylene C was assessed by X-ray diffraction (XRD) using an MPD X’Pert PRO powder diffractometer from PANalytical (Royston, UK), with a Cu Kα radiation source (*λ* = 1.540598 Å) equipped with a 1D X’Celerator detector. The XRD measurements were performed in the range of 10 to 65° (2*θ*) in the Bragg–Brentano configuration, with a scanning step size of 0.05° in 2*θ*. In situ XRD measurements were also performed using a non-ambient chamber (TC-Basic Modular System—Temperature Chambers for XRD, from Materials Research Instruments, MRI, which was acquired by Brucker Corporation, Billerica, MA, USA) adapted on the diffractometer within a temperature range between 25 °C and 200 °C to study the influence of temperature on the crystallinity of Parylene C. To assess the dielectric properties of Parylene C, MIM crossbar capacitors were fabricated on glass substrates. Aluminum electrodes (Plasmaterials Inc., Livermore, CA, USA) with different widths were deposited by e-beam evaporation (homemade system) and patterned through shadow masks forming the bottom and top contacts of MIM devices, where Parylene C layers with different thicknesses constitute the dielectric. An annealing step of 150 °C to 180 °C for 1 h was performed to mimic the processing conditions used for thin-film transistors within our group [29,42,43]. The electrical characterization of the MIM capacitors was performed using a semiconductor parameter analyzer (Keysight B1500A, from Keysight Technologies, Santa Rosa, CA, USA) connected to a probe station (Cascade EPS150 TRIAX from Microtron, Mechelen, Belgium), for measuring both the capacitance−voltage (*C-V*), in the range of −2 V to 2 V, and capacitance–frequency characteristics (*C-f*), in the range of 0.1 Hz to 1 MHz.

### 2.2. Parylene C-Based Devices

#### 2.2.1. Thin-Film Transistors (TFT): Device Production

Indium Gallium Zinc Oxide (IGZO) TFTs with Parylene C dielectrics were fabricated with a staggered bottom-gate structure with channel width and length of 320 μm and 20 μm, respectively. Patterning was performed using standard photolithography procedures and a Suss MA6 aligner (SUSS MicroTec, Garching, Germany). Sputtered layers were deposited by radio frequency (RF) magnetron sputtering in an AJA ATC-1300F system (AJA International Inc., North Scituate, MA, USA). For metallic gate, source, and drain electrodes, 60 nm films were sputtered from a metallic Mo target. Transparent electrodes were 180 nm films sputtered from a Gallium-doped Zinc Oxide (GZO) ceramic target (5:95 wt% Ga:ZnO from Alineason Materials Technology GmbH, Frankfurt am Main, Germany). The semiconductor was a 40 nm film sputtered from a 2-inch multicomponent ceramic target of IGZO (Indium:Gallium:Zinc Oxide in the proportion of 2:1:2 mol, also from Alineason). Parylene C was deposited as previously described, with thicknesses of 200 nm for the dielectric layer, 1 μm for the passivation, and 15 μm for use as a substrate layer (no adhesion promoter was used in this case). All sputtered layers were patterned by lift-off, while Parylene C vias on the dielectric layer were patterned using reactive ion etching (Phantom III RIE from Trion Technology, Tempe, AZ, USA) under an oxygen atmosphere, with 30 SCCM of O_2_ during 70 s at a pressure of 1000 mTorr and power of 25 W. Samples were annealed at 180 °C for 1 h in a hotplate and no intentional substrate heating was performed at any other process step.

#### 2.2.2. Thin-Film Transistors (TFT): Device Characterization

The electrical characterization of the TFTs was also performed using a semiconductor parameter analyzer (Keysight B1500A, from Keysight Technologies, Santa Rosa, CA, USA) connected to a probe station (Cascade EPS150 TRIAX from Microtron, Mechelen, Belgium).

#### 2.2.3. MIM Structures for Digital Microfluidic (DMF) Applications: Device Production

MIM structures were produced, consisting of a 2 μm layer of Parylene C, sandwiched between chromium bottom and top contacts. Figure 1 depicts this structure.

Glass substrates were cleaned with acetone and isopropyl alcohol (IPA) baths (15 min each, at 50 °C) under ultrasound actuation (model Sonorex Super—Bandelin electronic GmbH, Berlin, Germany), further rinsed with ultrapure water, and dried with a nitrogen gun. Chromium (also from Plasmaterials) bottom contacts (200 nm) were deposited by an in-house electron beam deposition system at 100 °C, through mechanical masks, over which the 2 μm layer of Parylene C was deposited according to the protocol described above. For single-layered Parylene C devices, a unique deposition of Parylene C was performed; however, for double-layered devices, two layers of 1 μm each were deposited, and all samples were nitrogen-blown between depositions. Finally, chromium top contacts were deposited as previously described, but at room temperature. The effective areas of the MIM structures are as follows: 0.0625 mm^2^, 0.25 mm^2^, 0.5625 mm^2^, and 1 mm^2^—see Figure 1.

#### 2.2.4. MIM Structures for Digital Microfluidic (DMF) Applications: Device Characterization

The dielectric layer (Parylene C) was characterized when exposed to electric and thermal stress, as required to perform DMF-based loop-mediated isothermal amplification (LAMP) of DNA [36] within the group. Thus, the MIM structures were exposed to three different settings: (1) constant temperature (62 °C); (2) constant AC actuation (20 V_RMS_); and (3) both stimuli simultaneously, which correspond to the DMF-LAMP operation parameters. All three conditions were applied to the devices for two hours, anticipating the use of our devices for the detection of low-concentration DNA targets. To apply temperature, a custom-built temperature unit was used, consisting of an ITO-covered glass substrate with titanium–gold contacts (heating element) and a PT100 temperature sensor, which was similar to the one previously reported by our group [36]. The temperature system differs only from the previously reported one in the location of the PT100 sensor, which was moved below the heating element, as shown by Figure 1d. Briefly, the temperature sensed by the PT100 (RS PRO PT100 class B 2 platinum chip—2 mm diameter, RS Components, Corby, UK) was reported to an Arduino board, and according to the temperature setpoint and the PID (proportional integral derivative) parameters, a MOSFET driver adjusted the current output to the heating element, consequently adjusting the temperature. As for the AC signal, a wave generator and an amplifier were used to apply a 20 V_RMS_ signal to the bottom contacts of the MIM structures, whereas the top contacts were grounded. This setup is part of our previously developed DMF platform and has been described elsewhere [36]. Lastly, for dual stimulation (temperature and AC signal), the same DMF platform was used to mimic on-chip LAMP conditions. All characterizations were performed using the Keysight B1500A semiconductor parameter analyzer and Cascade Microtech EPS 150 manual probe station. *C-V* measurements were performed for a voltage range of −5 V to +5 V (step of 0.5 V), followed by another voltage loop from +5 V to −5 V. Dielectric breakdown was then attempted, by applying a high voltage DC signal with a total range amplitude of 200 V. During this measurement, the leakage current was recorded for a step of 2 V (current–voltage or *I-V* measurements). Said measurements were performed for each sample, before and after applying the stimuli. Moreover, Parylene C is known to crystallize with temperature; therefore, a structural analysis was performed by XRD using the PANalytical X’Pert Pro diffractometer. Diffraction spectra were collected in continuous mode from 10° to 20° (2*θ*), for a step of 0.002°, with a generator voltage of 45 kV and a tube current of 40 mA.

#### 2.2.5. Digital Microfluidics: Device Production

DMF devices were produced as previously disclosed [36]. Briefly, glass substrates were cleaned by gentle rubbing with detergent, followed by acetone and IPA ultrasound baths at 50 °C for 15 min each. For the bottom plates, after photolithographic patterning of the electrode paths, a 200 nm layer of Cr was deposited by e-beam at 100 °C. Following lift-off, Parylene C was deposited in double layer as described above, and a PTFE (polytetrafluoroethylene) AF 1600 solution consisting of 0.6% *wt*/*wt* of PTFE AF 1600 in Fluorinert FC-40 (DuPont, Wilmington, DE, USA) was deposited over the Parylene C by spin-coating, achieving a final thickness of 50 nm. After depositing the PTFE solution, annealing followed at 160 °C for 10 min. The top plates consisted of glass coated with indium-tin-oxide (ITO), over which a layer of PTFE was also deposited. Spacing between the top and bottom plates (180 μm) was assured by polyimide tape, and the electrode area was sealed using nail polish and dried at 55 °C for 10 min. Lastly, 5 cSt silicone oil (Sigma-Aldrich, St. Louis, MO, USA) was used as a filler.

#### 2.2.6. Digital Microfluidics: Device Characterization

DMF devices were characterized in terms of droplet speed and the ability to withstand long LAMP reactions up to 2 h. For determination of droplet speed, videos of droplets moving back and forth from one electrode to an adjacent one were captured by a digital microscope (Handheld Digital Microscope Pro from Celestron, Torrence, CA, USA). Such videos were then used to determine the time required for both the head and the tail of droplets to start moving from an unenergized electrode to the adjacent energized electrode. Speed was determined by dividing the travelled distance by the time required to travel it. For long LAMP reactions, the *18S* gene was amplified on-chip following the mixing and fluorescence detection protocols disclosed previously [36]. Amplification was performed at a temperature setpoint of 61.5 °C for 2 h.

## 3. Results and Discussion

### 3.1. Parylene C Structural and Thermal Characterization

Parylene C is a semicrystalline polymer that has a monoclinic structure with a = 5.96 Å, b = 12.69 Å, c = 6.66 Å, and *β* = 135.28° unit cells [44,45,46]. It is a polymer with a relatively low glass transition, near 50 °C, and a melting point of 290 °C. Moreover, it has a thermal expansion coefficient of 35 ppm/°C. As such, when submitted to temperatures above 50 °C, Parylene C films increase their crystallinity and undergo a substantial lattice expansion that further results in a more compact structure after returning to room temperature. In most fabrication processes, as in photolithography, upon deposition of other layers or annealing treatments, the crystallization and expansion/contraction of the films will be induced. As such, the characterization of Parylene C structural properties, while submitted to temperature stimuli, is of extreme importance, since such stimuli might affect the overall electrical and mechanical properties of the final devices.

Figure 2a presents the diffractograms of Parylene C films with different thicknesses ranging from nearly 0.2 μm (thickness in the same order of magnitude as for dielectrics in transistors [12,25,26]) to nearly 4 μm (in the range for both encapsulation [28,29] and substrate [11,23] purposes). These films were not submitted to any thermal treatment, and therefore, they represent the as-polymerized state. The polymerization process for Parylene C is performed at room temperature, contrary to common polymers that polymerize under a thermal stimulus. All samples exhibit a single peak near 2*θ* = 13.8° related to the (020) plane of the Parylene C structure. The thicker the film, the higher the intensity detected, but no meaningful changes are visible on the peak position and shape.

To study the behavior of post-deposition thermal treatments (such as annealing steps for TFT production [12,29]), a sample of Parylene C with 240 nm thickness deposited on a Si wafer piece was heated from 25 °C to 200 °C (in heating steps of 25 °C), and data were collected in situ to follow the crystallization kinetics. For each step, the sample was kept at the respective temperature for 25 min, and consecutive scans were collected in a non-ambient XRD chamber. The sample was cooled down to room temperature after each temperature step so that the immediate effect of the temperature on the crystalline properties could be observed. The diffractograms obtained in this study are presented in Figure 2b, and the evolution of the respective d-spacing is presented in Figure 2c.

The initial peak position, located at 13.8°, represents a d-spacing of (6.411 ± 0.005) Å on the (020) plane of the monoclinic structure of Parylene C. As depicted in Figure 2b, there is no substantial change in the peak position or width until 50 °C. However, by increasing the temperature, there is a prominent shift to lower 2*θ* angles, indicating a lattice expansion followed by an increase in the intensity of the peaks and a decrease in their full width at half maximum (FWHM), confirming the crystallization of Parylene C. The data show not just the thermal lattice expansion but also the lattice contraction upon cooling to room temperature. In fact, the final state is much more compressed than the initial state, which is due to heating above the glass transition temperature. Such results clearly indicate that each thermal treatment applied to a Parylene C film will induce its crystallization and a huge lattice expansion and contraction (see Table 1). It is worth noting that when cooling down to room temperature, the lattice contraction observed will depend on the maximum temperature that the sample was submitted to. Moreover, the higher the temperature, the higher the contraction observed when the sample returns to room temperature. This phenomenon occurs because Parylene C was heated above its glass transition temperature under an oxidizing atmosphere. The crystallite size, or the domain size, was evaluated using Scherrer’s equation:(1)$$D=\frac{K\lambda }{\beta \cos\left(\theta \right)}$$
where *K* is the shape factor (set to 1), *D* is the apparent crystallite size, *λ* the wavelength of the radiation, and *θ* is the Bragg diffraction angle—corresponding to the (020) plane and *β* is the full width of the diffraction FWHM of the instrument-corrected line profile (not to be confused with the angle of the monoclinic structure of Parylene C mentioned previously, which is also denoted with *β*). Table 1 summarizes the main results obtained, including the overall thermal expansion and the crystallite dimensions upon each thermal annealing (measured at room temperature). The domain size has an almost three-fold increase upon the thermal treatment performed at 200 °C.

### 3.2. Electrical Properties of Parylene C

To study the electrical properties of Parylene C, MIM capacitors with different dielectric thicknesses and a 2.25 cm^2^ area were fabricated. Shadow masks, such as the ones shown in Figure 1c, were used to pattern electrodes with a width of 1.5 mm (an area of 2.25 mm^2^). The characterization of the films was performed through *C-V*, *C-f*, and *I-V* characteristics. From the *C-V* curves at 100 kHz, it was possible to determine the capacitance per unit area (*C/A*), the dielectric constant (*κ*) and the loss tangent (tan δ). Samples were further submitted to a maximum voltage of 100 V to assess their dielectric strengths—see Table 2. Please note that 100 V is the maximum voltage applied by the available equipment; therefore, the maximum electric (E) field tested and presented in Table 2 corresponds to the maximum field that we were able to test, and not to the strongest field that the material can withstand.

As can be observed, at 100 kHz, the dielectric constants of Parylene C are approximately 3 [7], which is close to the ones reported in other sources [7,47,48], and the loss tangent values are relatively low and in the same order of magnitude reported in the literature [48]. Furthermore, all MIM devices could withstand a DC voltage of 100 V without achieving breakdown. Applying 100 V_DC_ leads to large electric fields that all tested devices were able to withstand, ranging from 0.69 MV/cm for a thickness of 1.45 μm to 4.11 MV/cm for a thickness of 0.24 μm, which attests to the robustness of the MIM devices with Parylene C as a dielectric.

### 3.3. Parylene C-Powered Devices

#### 3.3.1. TFT Oxide Electronics with Parylene C

TFTs of amorphous oxide semiconductors are compatible with transparency and flexibility and can be integrated into non-conventional substrates, such as Parylene C, due to their low-temperature fabrication, making them an interesting alternative for electronics with/on Parylene C. Herein, we show IGZO TFTs using Parylene C as dielectric, substrate, and passivation layers. Devices employ 200 nm Parylene C as the dielectric layer and 1 μm Parylene C as the passivation layer. This passivation layer has been shown to decrease the aging of similar oxide TFTs (providing excellent performance after 7 months of storage in the dark) and to increase their stability under both bias and illumination stress [29]. Besides devices with metallic electrodes fabricated on Corning substrates, a fully transparent and flexible device employing GZO transparent electrodes and fabricated on a 20 μm Parylene C layer is shown. Device schematics are shown in Figure 3.

Linear and saturation transfer curves for both types of devices are shown in Figure 4. Both device types present steep transfer curves, with subthreshold slopes of 0.26 V/dec, and negligible gate currents, showing that the Parylene C dielectric layer is providing proper leakage insulation for the applied biasing range. While a high mobility of 16 cm^2^V^−1^s^−1^ is shown for devices employing the Mo electrodes, it is clear that using GZO electrodes poses some contact resistance, causing the field effect mobility (μ_FE_) to be slightly lower than that in saturation (μ_Sat_). Regardless, these devices present a decent performance, showing that fully transparent IGZO TFTs devices can be fabricated on Parylene C layers and that Parylene C as a dielectric can provide a good channel interface with oxide semiconductors yielding performances comparable to employing oxide dielectrics. While other a-IGZO transistors with Parylene C as dielectric were also reported to have performances comparable to those of oxide dielectrics, the metrics for these devices (μ_FE_ = 12.9 cm^2^V^−1^s^−1^ and subthreshold slope of 0.651 V/dec) are not as good as those here presented, which may be due to the use of lower annealing temperatures [14]. Other transistors have been developed using Parylene C as a dielectric [12,25] or even in dual functionality, with Parylene C acting as a dielectric and a substrate [11]. Transistors with Parylene C gate dielectrics either present similar properties to the ones presented herein, but require additional materials for the dielectric layer (higher fabrication complexity) [25]; or present poorer performance in terms of carrier mobility, for example [12]. Transistors with Parylene C acting as a dielectric and as a substrate [11] may present better performance, with very high mobility, low leakage, and low gate operating voltage; however, they also rely on 2D materials, increasing fabrication complexity.

#### 3.3.2. MIM Structures Based on Parylene C for Mimicking Digital Microfluidic (DMF) Stress Conditions: Combined Thermal-and Electrically Stressed Devices

The dielectric layer is a crucial component in DMF devices, preventing electrolysis when an electric field is applied to move liquid droplets [49]. Within our group, DMF devices are used for isothermal nucleic acid amplification [36,37,38], which requires heating of the devices and applying voltage to reaction electrodes for a period of time, around 60 min to 90 min. Herein, we study the effect of heating and electric stress on the Parylene C dielectric layer by applying temperature and an AC signal to the MIM structures (shown in Figure 1), similar to the ones used for extracting the electric properties of Parylene and for DMF nucleic acid amplification. Higher electrical strengths can be attained by stacking Parylene layers; thus, a comparison between a single 2 μm-thick Parylene layer and a double-layered structure with 1 μm thickness per layer is shown. More specifically, we study the influence of applying 62 °C and/or 20 V_RMS_ to the MIM structures for a period of 2 h. The temperature setpoint was rounded to 62 °C, in comparison to previous assays, to prevent maximum temperature error [36], and 20 V_RMS_ was the voltage signal applied to droplets during the reaction: high enough to prevent droplets from moving, but low enough to prevent device degradation. A period of 2 h was chosen considering that this was the maximum reaction time attempted by our research group. As described in the Materials and Methods section, *C-V* measurements were performed on each sample before and after applying the stimulus (temperature and/or AC signal). Briefly, for each tested condition, a minimum of three samples were tested, and the results obtained for the capacitance were averaged and normalized to the corresponding area (capacitance per unit area, *C/A*). The *C/A* obtained after applying the stimulus was further normalized to the *C/A* obtained before applying the stimulus, according to Equation (2):(2)$$\Delta \frac{C}{A}=\frac{\left(\frac{C}{A}\;post\;stimulus\right)-\left(\frac{C}{A}\;pre\;stimulus\right)}{\frac{C}{A}\;pre\;stimulus}\times 100$$

For simplification, the capacitance per unit area will from now on be referred to as *C/A*, and the relative variation of the capacitance per unit area, obtained through Equation (2), will be denoted as Δ*C/A*. Figure 5 illustrates the average Δ*C/A*. For pre-stimulus device characterization in terms of capacitance, relative permittivity, and loss tangent, please refer to Appendix A and Appendix A, respectively.

Figure 5a illustrates the relative variation of the capacitance per unit area (Δ*C/A*) with temperature, for all tested areas, with Parylene C deposited as a single or double layer. Overall, the *C/A* decreases with temperature for both single- and double-layered Parylene C (as inferred by the negative Δ*C/A*), and larger areas seem to promote a greater variation, despite the area normalization of the data. Moreover, double-layered Parylene C reveals a slightly larger decrease in *C/A* as the MIM area increases. This could be due to the interface between both Parylene C layers, which represents a discontinuity in the thin films and is possibly more prone to defect propagation, leading to small decreases in capacitance. Nevertheless, the maximum average decrease in *C/A* in all samples is about −11%, which does not represent a meaningful variation. However, Perylene C is known to crystallize with temperature, and an increase in the crystallization of the polymer could explain the general decrease in *C/A* with this stimulus. Thus, XRD analysis was performed on pristine samples without chromium electrodes. Said samples were subjected to the same temperature stimulus, and the obtained diffractograms are presented in Figure 6. This figure also includes diffractograms for the dielectric layer before and after the annealing step required for PTFE curing during the fabrication of DMF devices (see Section 2.2.5).

Figure 6a shows that the un-annealed dielectric layer presents XRD intensity peaks at 13.85° and 13.84° for double- and single-layered Parylene C, respectively. Data are in accordance with the literature results reported for Parylene C only [26,50,51], which suggests that for an XRD analysis, the PTFE layer is not substantial. This was expected considering the extremely thin nature of the PTFE coating (50 nm), which falls out of the operation range of the diffractometer. Furthermore, the X-ray diffraction peak detected at around 2*θ* = 14° has been associated with the (020) plane of the monoclinic unit cell for Parylene C only [44,52]. The peak for double-layered Parylene C is less intense than its single-layered counterpart (see Figure 6 and Table 3), which could be due to the interface between both layers. This interface produces a gap that could be translated to an apparent loss in crystallinity, hence the lower diffraction peak. The full width at half maximum (FWHM) is also slightly larger for double-layered Parylene C, with or without curing, supporting the apparent loss of crystallinity for double-layer structures. After being subjected to curing at 160 °C, the samples present a considerable rise in peak intensities, suggesting an increase in the crystallinity of the polymer for both single- and double-layered Parylene C. Similarly to what is observed in the as-deposited samples, the double-layer structure post-curing presents a lower peak than for single-layer structures, again suggesting a seeming loss in crystallinity. Accordingly, the FWHM also decreased after curing, indicating an increase in crystallinity as well. A right shift in the peak position (higher 2*θ*) is also observed for both structures after curing, which is usually an indication of a contraction of the lattice. Such an observation is reflected by the slight decrease in spacing between atomic layers, d-spacing (Table 3). Finally, the crystallite size is also increased after the curing process.

Figure 6b and Table 4 compare the cured dielectric layer before and after being subjected to the temperature stimulus, i.e., 2 h at 62 °C. The pre- and post-temperature stimulus XRD spectra overlap, indicates that low-temperature heating does not critically affect the crystallinity of the dielectric lattice, despite the duration of the stimulus. Therefore, the decrease in the capacitance of the dielectric layer after applying the temperature stimulus may not be entirely related to differences in crystallization. Other factors could be triggering the decrease in capacitance, such as a decrease in the dielectric constant, because of the lower polarization of the polymer molecules due to the increase in thermal energy.

Figure 5b illustrates the variation of the capacitance with AC actuation. For single-layered Parylene C, there is an overall decrease in the capacitance of the dielectric layer after applying the stimulus; however, the variation in capacitance becomes less pronounced as the area of the MIM structures increases. Regarding double-layered Parylene C, the capacitance increases after applying the stimulus. One possible explanation is the disorientation of the polymer molecules with the AC field, which could contribute to the increment of the capacitance. This phenomenon may be more expressive in double-layered Parylene C due to the additional interface between the two layers, which could facilitate molecule rearrangement, in comparison to its single-layer counterpart. Overall, both capacitance variations (for single- and double-layered Parylene C) tend to decrease as the area increases. This observation could be due to the decrease in the influence of the parasitic capacitances that may occur in the overlapping electrode region.

It should be noted that the maximum absolute variation in capacitance is around 3%, which is a good indication of the stability of the dielectric layer when subjected to an AC signal stimulus, as opposed to thermal stimulation, which may lead to three times higher variations in capacitance. This observation is in accordance with the literature, where AC signals are advised for droplet motion in electrowetting-based devices for a number of reasons, including the following: (1) contact angle saturation occurs for higher voltages [53]; (2) prevention of charge injection on the dielectric layer [54]; (3) reduction of the biofouling phenomenon (adhesion of biomolecules to the dielectric layer) [55]; and (4) improvement of mixing within the droplet [56].

Lastly, Figure 5c illustrates the variation in the capacitance when a dual stimulus is applied, merging both the temperature and AC signal stimuli. For both single- and double-layered Parylene C, the capacitance decreased after applying the stimuli to the MIM structures. Nevertheless, this is more expressive for single-layered Parylene C, even though at a very small scale, with a maximum absolute variation of approximately 4%. Indeed, single-layered Parylene C presents a more stable behavior regarding capacitance variations for individual stimuli, consistently close to 0%, with a maximum variation around 3%. This tendency is only reversed when both stimuli are applied simultaneously, and in this case, the double-layered Parylene C clearly presents a higher stability, with a maximum absolute capacitance variation that does not reach 2%, indicating that this approach could be a safer choice for electrowetting applications such as DMF. Interestingly, for the double layer, there seems to be an approximate sum of the capacitance variance for each individual stimulus. The maximum absolute variation in capacitance is about 1.4%, which is an intermediate value between −11% measured for the thermal-only stimulus and +3% measured for the AC signal-only stimulus. Nonetheless, this is a small variation percentage, which suggests that the dielectric layer under test is perfectly capable of withstanding heating and an AC electric field over a period of two hours.

The maximum leakage current (total range of 200 V) was measured to further characterize the MIM structures produced. Please note that, in this case, the data were not normalized, since the absolute differences in the leakage current are relevant to understanding whether devices are close to reaching dielectric breakdown. Figure 7 illustrates the maximum leakage current measured for all stimuli applied to the MIM devices.

Generally, the maximum leakage current measured increases with the area of the MIM capacitors. This is expected since, for larger areas, there is a higher probability of reaching defect-rich spots that promote the current flow through the dielectric layer. Considering the temperature stimulus (Figure 7a), for both single- and double-layered Parylene C, the maximum leakage current is low, below 0.4 nA, and does not vary considerably after applying the stimulus. Such an observation suggests that the decrease in capacitance (namely for double-layered Parylene C) is mostly due to molecule rearrangements within the polymer lattice and not to the occurrence of current pathways through the dielectric layer. As for the AC signal stimulus (Figure 7b), the maximum leakage current is consistently below 0.6 nA, indicating good isolation of said layer. Nevertheless, there appears to be a constant increase in the maximum leakage current after applying the AC stimulus for both single- and double-layered Parylene C. This is consistent with subjecting the dielectric layer to an electric field, which may remove some electrons from their positions within the molecular structure, thus creating paths for current flow through the insulation layer. Nevertheless, this effect remains negligible, considering that the leakage currents are acceptable for dielectrics. Lastly, for two stimuli applied to the dielectric layer (Figure 7c), the leakage current becomes considerable in the case of single-layered Parylene C, even though the current does not reach high levels, with a maximum average below 3 nA. However, this finding suggests that when an AC signal and temperature are simultaneously applied to a dielectric layer comprising single-layered Parylene C, the leakage current could become a relevant mechanism for the decrease in capacitance. It is also noteworthy that double-layered Parylene C remains on a low leakage current regime after the dual stimulus, which again suggests that this could be a safer option for electrowetting-based DMF devices than single-layered Parylene C. Of course, it should be noted that leakage currents were measured by applying a much higher voltage (200 V total range) to the insulator than for measuring the capacitance (10 V total range); therefore, further studies would be required to pinpoint the exact weight of this mechanism on the final reduction in the capacitance of the MIM structures.

Another interesting alternative to increasing the overall robustness of the dielectric layer in DMF devices is the combination of dielectric materials instead of performing multiple depositions of the same material. This approach has been used for reductions in the actuation voltage in electrowetting-based devices [57,58,59]; however, to the best of our knowledge, there have been no studies focusing on the characterization of a double-material dielectric layer for DMF bio-applications. We have begun to exploit double-material dielectric layers composed of tantalum pentoxide and Parylene C and have achieved promising results (see Appendix A).

#### 3.3.3. Digital Microfluidic Devices Based on Parylene C

As previously mentioned, DMF greatly benefits from a dielectric, conformable, and biocompatible polymer, such as Parylene C. Indeed, a considerable portion of DMF devices for nucleic acid amplification resort to Parylene C as the dielectric layer [31,60]. Nevertheless, even though Parylene C is hydrophobic in nature, DMF devices commonly resort to an additional layer to increase hydrophobicity (and consequently, increase the contact angle between liquid and the device surface). Considering that PTFE AF 1600 is greatly used as a hydrophobic layer in DMF devices, namely the ones produced within our group, we opted to characterize the performance of a dielectric layer composed of Parylene C and the hydrophobic layer. Figure 8 illustrates the latest DMF device produced by our group for nucleic acid amplification purposes [36]. Briefly, the devices are comprised of bottom and top plates and are separated by a spacer. The bottom plate includes metallic electrodes that may be activated or deactivated (by applying or removing AC signals), thus enabling the motion of small sample droplets from an electrode to an adjacent one. To prevent electrolysis of the aqueous droplet, a dielectric layer (Parylene C) is deposited between the electrodes and the droplets, and a hydrophobic layer is further applied to increase the droplet contact angle and facilitate motion. Appendix A includes SEM (scanning electrode microscopy) images of the bottom plate of our DMF devices. The top plate also includes a hydrophobic layer and a conductive layer to which the ground signal is applied.

As stated in the Introduction, Parylene C is one of the most common dielectric materials used in DMF. Indeed, Parylene C presents a very high dielectric strength (approximately 220.5 V/μm [7]), thus enabling droplet motion at relatively high actuation voltages without the risk of dielectric breakdown. Of course, the choice of Parylene C thickness is paramount for the device operation, and for our DMF devices, an optimal thickness of 2 μm was chosen. This thickness enables a large voltage operation range (breakdown will theoretically occur at around 441 V). Figure 9 represents the behavior of the dielectric layer according to the thickness of Parylene C, in terms of breakdown voltage and actuation voltage required for a 20° shift in contact angle (from 120° to 100°) in electrowetting-on-dielectric (EWOD) conditions. Please note that an initial angle of 120° was chosen considering that it is the initial contact angle with the hydrophobic PTFE layer. The EWOD actuation voltages per Parylene C thickness were determined in accordance with the Lippmann–Young equation represented below [39]:(3)$$\cos\theta =\cos\theta_{0}+\frac{1}{\gamma_{LG}}\times \frac{CV^{2}}{2}$$
where *θ_0_* and *θ* are the initial and final contact angles, respectively, *γ_LG_* is the liquid–gas surface tension (in this case, the surface tension between the liquid droplet and the filler silicone oil), *C* is the capacitance per unit area, and *V* is the applied voltage. *C* can be further developed as follows:(4)$$C=\frac{\varepsilon_{0}\varepsilon_{R}}{d}$$
where *ε*_0_ and *ε_R_* are the permittivity of the vacuum and the relative permittivity of the considered material, respectively, and *d* is the thickness of the dielectric material.

Note that two EWOD conditions are represented: 1) the voltage required to achieve a 20° shift in contact angle for a Parylene C-only dielectric layer (“EWOD ParC”) with varying thickness, and 2) the voltage required to achieve a 20° shift in contact angle for a dielectric layer comprised of a 50 nm layer of the hydrophobic layer plus a Parylene C layer with varying thickness (“EWOD ParC and 50 nm PTFE”). Considering that the hydrophobic layer is based on a 0.6% *wt*/*wt* solution of PTFE AF 1600 (*ε_R_* = 1.93 [61]) in Fluorinert FC-40 (*ε_R_* = 1.9 [62]), a relative permittivity of 1.9 was considered in calculations with the hydrophobic layer. Please note that Parylene C and the hydrophobic layer form parallel capacitors, with similar dielectric constants; therefore, the capacitance of a 50 nm layer of the PTFE in FC-40 solution will be negligible in comparison to the 2 μm layer of Parylene C (as confirmed by Figure 9). This also attests that the results obtained for MIM structures may also be associated with the dielectric layer of DMF devices, comprised of Parylene C and the hydrophobic layer, with minor error.

Figure 9 clearly illustrates the great advantage of using a dielectric layer featuring Parylene C, with a high dielectric strength, which allows for a wide range of EWOD actuation voltages, suitable even for low-conductivity solutions that may require higher actuation voltages. Additionally, adding a hydrophobic layer results in little variation in the actuation voltages required, which is also a good indicator for the suitability of the chosen PTFE layer to the Parylene C dielectric. The hydrophobic layer is also a key element in electrowetting-based DMF devices since it reduces the surface energy in contact with the droplets. Lower surface energy leads to less friction between the droplet and the surface and, consequently, droplet motion at lower voltages [49,63]. DMF devices relying on Parylene C and PTFE as dielectric and hydrophobic layers (respectively) have been adopted for several NAA methodologies, which is a good indicator of the suitability of such layers for NAA applications. Such methodologies include the gold standard amplification technique, PCR (polymerase chain reaction) [64], as well as isothermal methodologies such as RCA (rolling circle amplification) [65] or LAMP [37]. Figure 10 presents some features of DMF devices produced herein, with double-layered Parylene C and a PTFE-based hydrophobic layer.

As illustrated by Figure 10a, the great advantage of using double-layered Parylene C on DMF devices is the possibility of performing longer isothermal nucleic acid amplification reactions (in this case, LAMP reactions) without device degradation. Longer reactions may be required for very low-concentration targets or for standard concentration targets, which are more difficult to amplify due to a number of factors such as sample complexity or pH, among others [66,67]. Moreover, double-layered Parylene C allows for the application of a higher amplitude voltage signal without device degradation, which in turn results in higher droplet motion speeds (Figure 10b). This increase in droplet motion speed can be relevant to preventing reagent degradation (e.g., photobleaching during dislocation to reaction sites in the case of DMF-LAMP) or other undesirable effects in time-sensitive reactions. Figure 10c demonstrates that applying 50 V_RMS_ to droplets instead of 40 V_RMS_ leads to a higher coverage of the electrode due to the lower final contact angle (Equation (3)), which may be relevant for applications where a fixed area is required (e.g., impedance measurements). Lastly, Figure 10d illustrates the droplet motion process from one electrode to an adjacent one in a DMF device comprising double-layered Parylene C, which is as straightforward as for devices with single-layered Parylene C (previously reported by our group [37]).

## 4. Conclusions

In this work, we investigated Parylene C in terms of its structural, thermal, and electrical behavior, and further studied electronic devices based on this polymer, namely thin-film transistors, metal–insulator–metal structures, and digital microfluidic platforms.

A structural characterization of Parylene C was performed by XRD, either at room temperature or with heating at several temperatures. An electrical characterization of the polymer was also performed to determine its capacitance per unit area, dielectric constant, and loss tangent. Briefly, thermal treatments on Parylene C led to crystallization of the lattice, as confirmed by increasing left-shifts of the 2*θ* angles as the temperature increased, accompanied by higher and narrower diffraction peaks (smaller d-spacings). As soon as Parylene C was exposed to temperatures higher than the glass transition temperature, the polymer expanded and further contracted upon cooling down to room temperature, stabilizing in a more compressed structure. This behavior is dependent on the temperature applied, and the higher the treatment temperature, the higher the expansion and corresponding lattice contraction. This phenomenon is extremely important to consider in devices based on Parylene C layers since the dynamics of other layers may not sustain such expansion/contraction cycles. Regarding the electrical behavior at room temperature, Parylene C exhibited decreasing capacitance per unit area with thickness increase, yet with low losses and dielectric constants close to those specified in the literature, attesting to the good quality of the deposited thin films.

Transistors were produced with Parylene C in double- or triple-functionality, i.e., as a dielectric and passivation layer or as a dielectric, passivation layer, and substrate. Such devices were electrically characterized through transfer curves and leakage currents. Regarding transistors produced with Parylene C as a dielectric and substrate, two types of devices were tested: fully transparent devices with GZO electrodes and non-transparent devices with Mo electrodes. Both device types showed low leakage currents, suggesting that Parylene C is a good fit as a TFT dielectric. Reasonable mobilities were determined for both device types; however, the mobility for devices with GZO contacts (transparent devices) is lower than for Mo contacts, which implies a higher contact resistance for GZO. Overall, devices present decent performance and attest to Parylene C forming a good interface with oxide semiconductors.

Lastly, to evaluate the influence of DMF-LAMP reaction parameters on the dielectric Parylene C layer, MIM structures based on single- and double-layered Parylene C were produced and subjected to the same temperature and AC signal actuation used for the referred reactions. When both stimuli are simultaneously applied, capacitance per unit area consistently decreases. However, there seems to be a cancellation of the effect of each individual stimulus on the dielectric layer for double-layered Parylene C, which presents a lower overall capacitance variation. This suggests that the double-layer approach is a safe option for preventing dielectric breakdown. Furthermore, DMF devices produced with double-layered Parylene C allow for faster droplet motion and enable longer reactions without device degradation. Of course, for electrowetting applications, there is still an important factor to consider: the influence of the liquid placed on the dielectric layer. Several studies have been conducted to evaluate the degradation of Parylene C while in contact with aqueous solutions [68,69,70,71,72], with and without the application of voltage. Specifically concerning EWOD-based DMF, applying a strong electric field between the top and bottom plates of the DMF devices will promote penetration of the liquid within such layers, especially if defects are present [72]. This is particularly relevant for the triple contact line connecting the droplet, the bottom plate materials, and the filler medium, where the electric field is stronger [73,74]. One study by Papathanasiou et al. [73] has suggested an increase in the magnitude of the electric field up to ten times at the triple contact line. Following this trend, we envision a thorough study specifically oriented toward DMF bio-applications, involving a wide range of biological buffers, as well as a comparison between strategies for degradation prevention (e.g., double-layered Parylene C or the use of two dielectrics).

## Figures and Tables

**Figure 1 polymers-15-02277-f001:**
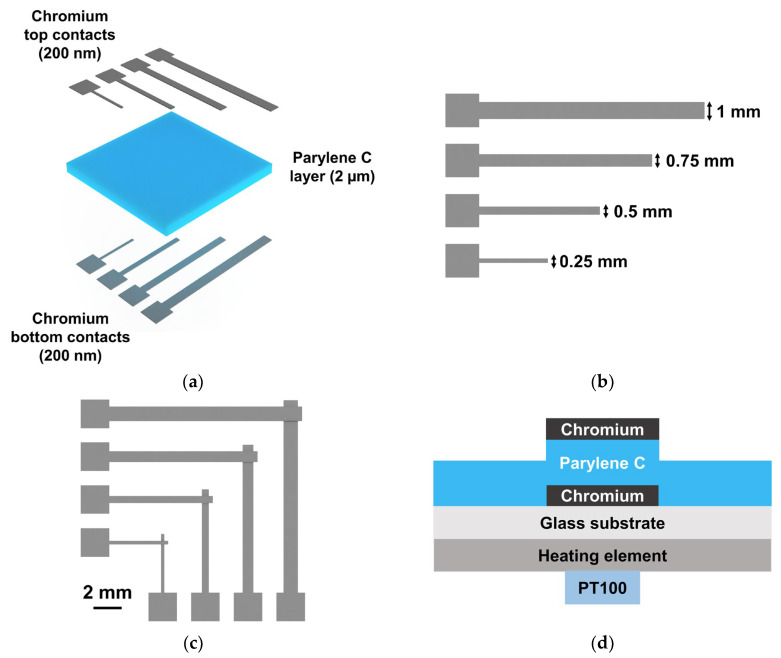
Schematic representation of the MIM structures produced for testing the Parylene C dielectric layer. (**a**) Three-dimensional view of all layers comprising the MIM structures, as well as respective thicknesses. Image not to scale. (**b**) Depiction of chromium contacts and respective lateral dimensions. (**c**) Overlap of chromium contacts, evidencing the area corresponding to the MIM capacitors (the Parylene C layer was omitted for better visualization). (**d**) Cross-section of a MIM structure placed over the temperature setup. Image not to scale.

**Figure 2 polymers-15-02277-f002:**
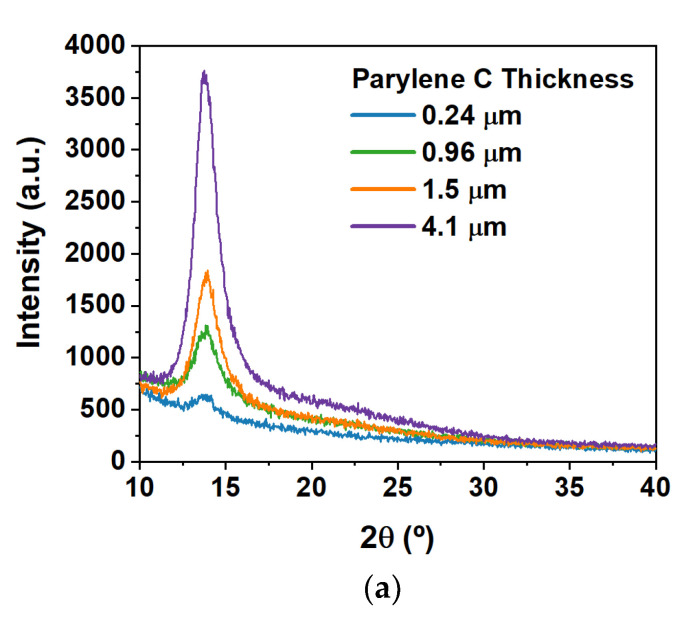
Crystalline parameters obtained for Parylene C thin films with several thicknesses at room temperature (**a**), and subjected to temperature treatments for a fixed thickness of 240 nm (**b**,**c**). (**a**) X-ray diffractograms obtained for Parylene C thin films with multiple thicknesses (0.24 μm, 0.96 μm, 1.5 μm, and 4.1 μm). The diffractograms were acquired with a Cu Kα radiation source (*λ* = 1.540598 Å) for a range of 10 to 40° (2*θ*) in the Bragg–Brentano configuration, and a scanning step size of 0.05°. (**b**) X-ray diffractograms obtained for a Parylene C thin film subjected to multiple heating steps. The full lines represent measurements performed with heating in situ at different temperatures (heating steps)—for each heating step, several diffractograms were collected. The dashed lines represent diffractograms obtained after cooling down to room temperature (RT) for each heating step, respecting the same color code (e.g., dark green full lines represent diffractograms collected at 125 °C, whereas dark green dashed lines represent diffractograms collected at RT after cooldown from 125 °C, and the temperature is afterwards raised to 150 °C for the next heating step). The inset represents the diffractogram for the first temperature setting (30 °C), as well as post-heating diffractograms for 50 °C and 75 °C, which are not visible in the larger graphic. Measurements were performed in the Bragg–Brentano configuration, with a scanning step size of 0.05°. (**c**) d-spacings were determined for all stages of heating. Data points above the red dashed line (blue dots) represent d-spacings determined for each heating step (horizontal axis), whereas data points below the red dashed line (green squares) represent d-spacings determined at RT after cooldown from each temperature step. The arrow indicates the largest difference in d-spacing.

**Figure 3 polymers-15-02277-f003:**
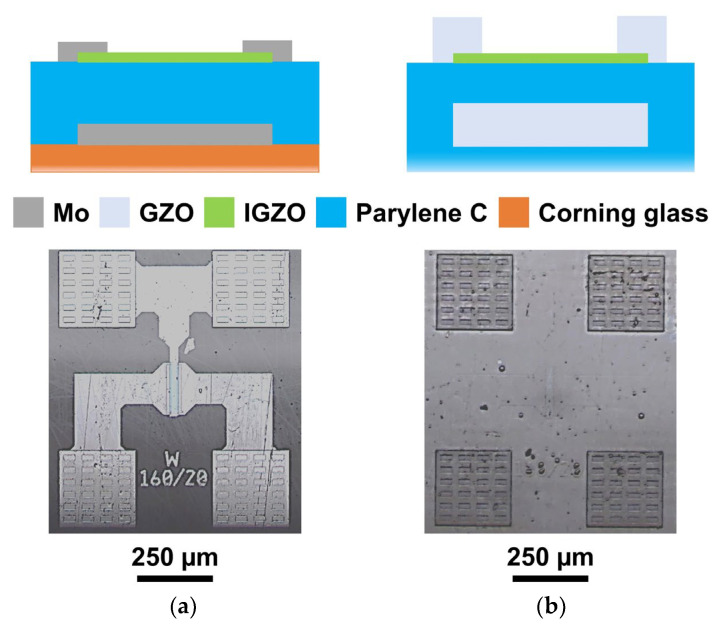
Schematics (top, not to scale) and micrographs (bottom) of IGZO TFTs with Parylene C as a multifunctional material, operating as a dielectric, passivation layer, and substrate, with metallic (Mo) electrodes (**a**) and fully transparent (GZO) electrodes (**b**).

**Figure 4 polymers-15-02277-f004:**
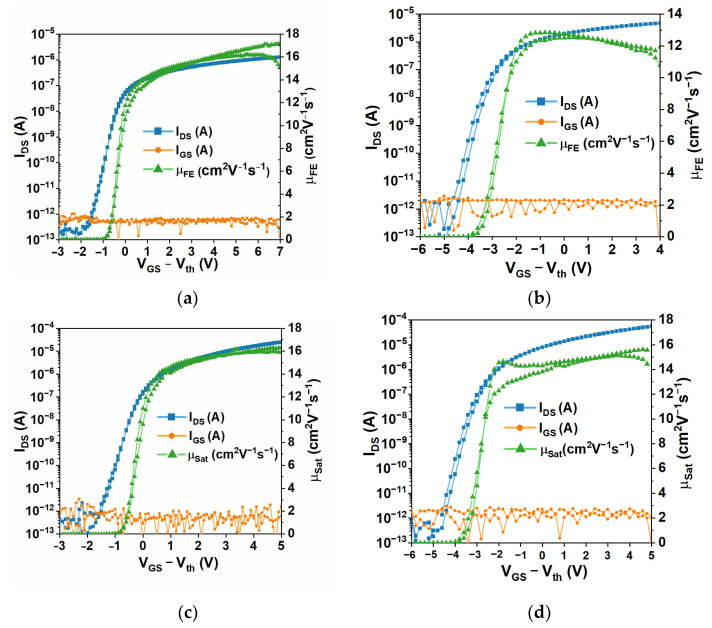
Characteristics of the IGZO TFTs produced with Parylene C as a multifunctional material, operating as a dielectric, passivation layer, and substrate: (**a**) linear transfer curves for TFTs with metallic (Mo) electrodes; (**b**) linear transfer curves for TFTs with fully transparent (GZO) electrodes; (**c**) saturation transfer curves for TFTs with metallic (Mo) electrodes; and (**d**) saturation transfer curves for TFTs with fully transparent (GZO) electrodes.

**Figure 5 polymers-15-02277-f005:**
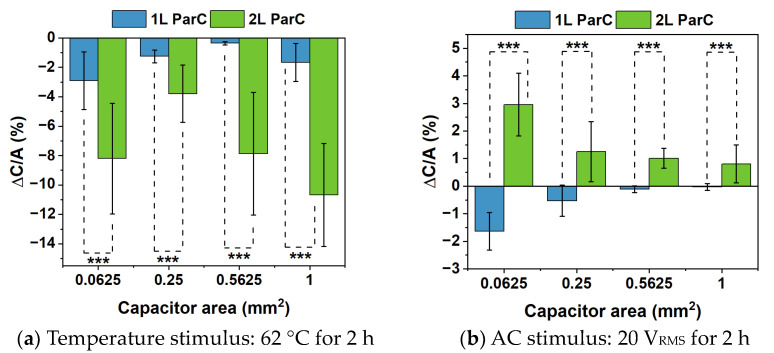
Δ*C/A* for the dielectric layer (Parylene C and PTFE) with Parylene C (ParC) deposited as a single (blue bars) or double layer (green bars), for four different MIM capacitor areas (considering that not all data follow a normal distribution (likely due to the measurement of capacitance in loop —see Materials and Methods), Kruskal–Wallis H Test with a significance level of 0.05 was performed to evaluate the differences between single- and double-layered Parylene C for each condition): 0.0625 mm^2^, 0.25 mm^2^, 0.5625 mm^2^, and 1 mm^2^. (**a**) Δ*C/A* with temperature (62 °C applied for 2 h). (**b**) Δ*C/A* with AC signal (20 V_RMS_ applied for 2 h). (**c**) Δ*C/A* with both stimuli (62 °C and 20 V_RMS_ applied for 2 h). Error bars correspond to the standard error. For (**a**,**b**), all capacitor areas demonstrate statistical differences between single- and double-layered Parylene C; *** represents *p* values ≤ 0.001. For (**c**), there are no statistical differences for 0.0625 mm^2^ and 0.25 mm^2^, whereas ** represents statistical difference, with *p* values ≤ 0.01.

**Figure 6 polymers-15-02277-f006:**
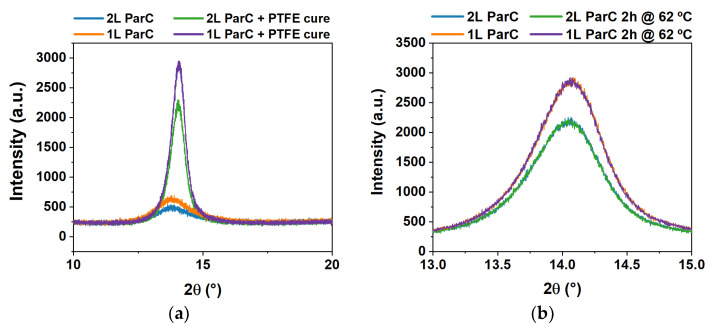
XRD diffractograms for the dielectric layer comparing all steps in the fabrication process involving temperature, with Parylene C deposited as a single (“1 L ParC”) or double (“2 L ParC”) layer. (**a**) Diffractograms prior (“ParC”) and following PTFE curing at 160 °C for 10 min (“ParC and PTFE cure”). (**b**) Diffractograms for the dielectric layer after PTFE curing, prior and following the temperature stimulus, i.e., heating at 62 °C for 2 h (“2 h at 62 °C”).

**Figure 7 polymers-15-02277-f007:**
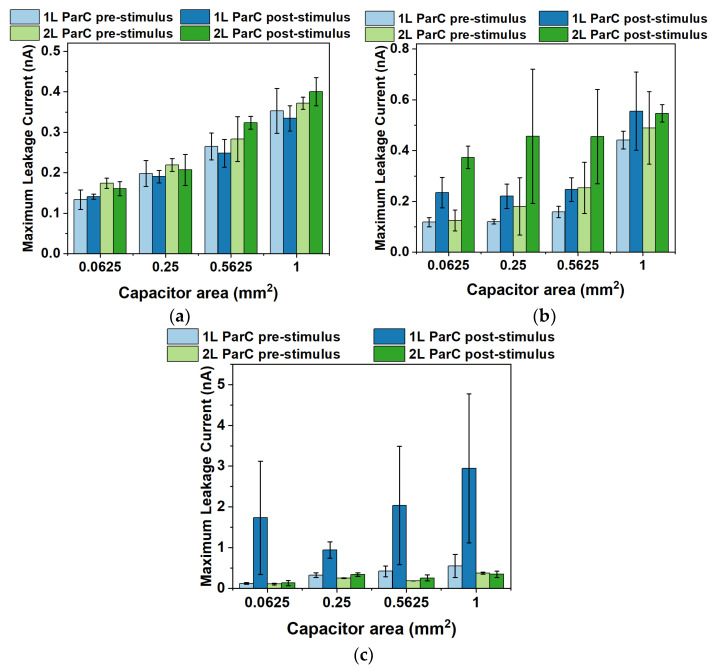
Maximum leakage current for the dielectric layer (Parylene C and PTFE) with Parylene C (ParC) deposited as a single (“1 L”) or double layer (“2 L”) for four different MIM capacitor areas. Leakage measurements were performed before (“pre-stimulus”) and after (“post-stimulus”) applying the stimulus. (**a**) Maximum leakage current measured for the temperature stimulus (62 °C applied for 2 h). (**b**) Maximum leakage current measured for the AC stimulus (20 V_RMS_ applied for 2 h). (**c**) Maximum leakage current measured for both stimuli (62 °C and 20 V_RMS_ applied for 2 h). Error bars correspond to the standard error.

**Figure 8 polymers-15-02277-f008:**
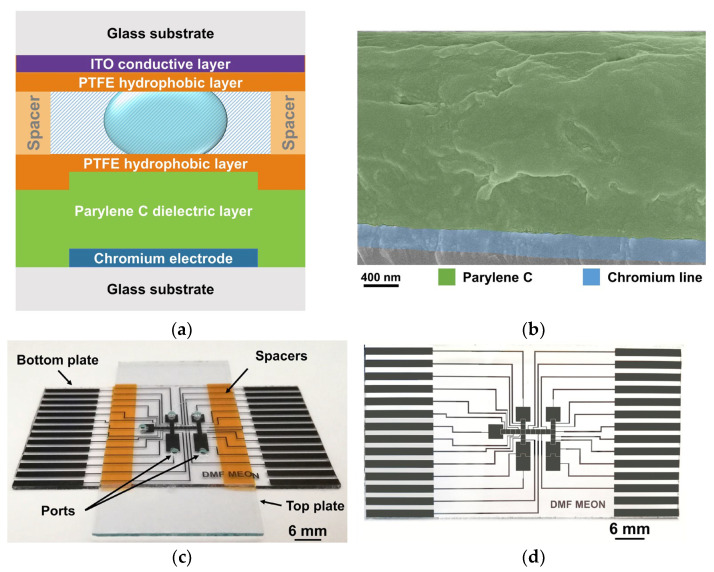
Example of a DMF device with Parylene C as a dielectric layer. (**a**) Schematic representation of a cross-section of a DMF device, where both top and bottom plates are represented, separated by spacers. Parylene C is deposited over chromium electrodes (where AC signals are applied to enable droplet motion), thus effectively preventing electrolysis. (**b**) SEM cross-section image of the bottom plate of a DMF device, highlighting the Parylene C dielectric (green filter) and a chromium connector line layer (blue filter). An image without such filters may be consulted in Appendix A. (**c**) Photograph of the assembled DMF device with bottom and top plates separated by 3 strips of polyimide tape. The top plate includes access ports that enable sample insertion and withdrawal. (**d**) Photograph of the bottom plate of a DMF device, evidencing the electrodes and connection lines.

**Figure 9 polymers-15-02277-f009:**
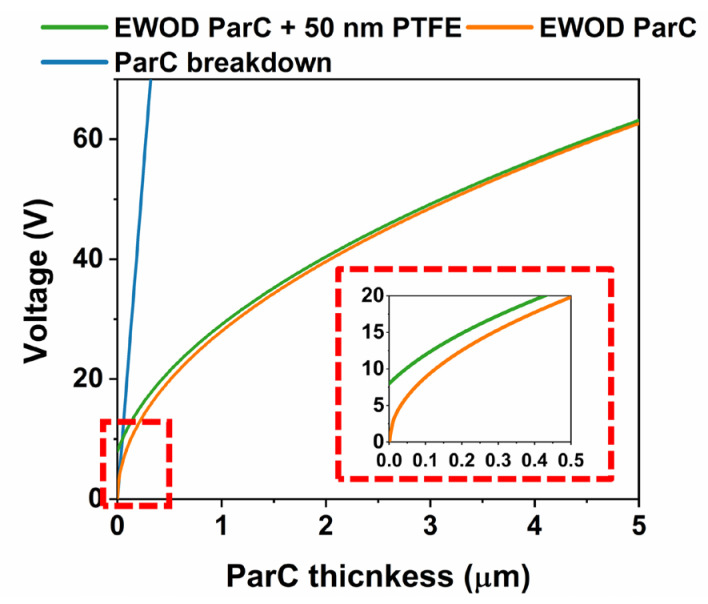
Behavior of Parylene C with varying thickness in terms of breakdown voltage (“ParC breakdown”), and voltage required to induce a contact angle shift of 20° in EWOD conditions. For the last case, two situations are represented: a dielectric layer comprised of Parylene C only (“EWOD ParC”) with varying thickness, and a dielectric layer comprised of 50 nm of PTFE plus a Parylene C layer with varying thickness (“EWOD ParC and 50 nm PTFE”). The inset shows the small region where the difference in required actuation voltage for Parylene C only and Parylene C and PTFE is more relevant. The breakdown voltage line was omitted for better visualization. Please note that for the breakdown voltage, only Parylene C was considered, since the breakdown voltage is considerably high.

**Figure 10 polymers-15-02277-f010:**
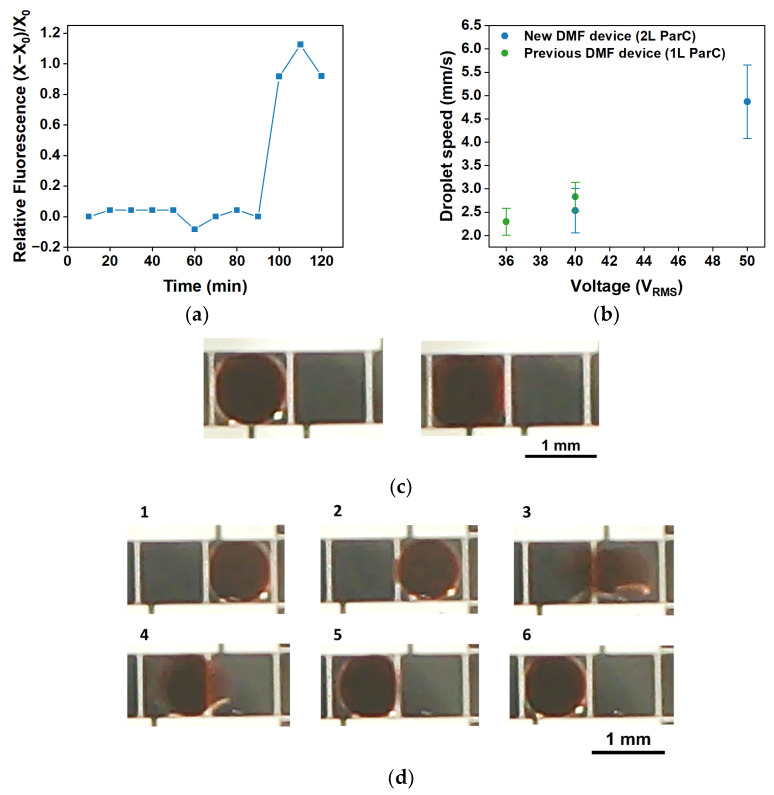
Aspects of DMF devices based on double-layered Parylene C. (**a**) Example of a long LAMP reaction enabled by double-layered Parylene C DMF devices, where target amplification occurs only from 90 min onwards. (**b**) Droplet speed achieved on the new DMF devices, featuring Parylene C deposited in double layer (“2 L ParC”), in comparison to droplet speed achieved on previous DMF devices, with single-layered Parylene C (“1 L ParC”). (**c**) Shape acquired by a droplet under a 40 V_RMS_ (left) or 50 V_RMS_ (right) signal. (**d**) Sequence of video frames showing droplet motion on a DMF device based on double-layered Parylene C (40 V_RMS_ applied to the droplet).

**Table 1 polymers-15-02277-t001:** Peak parameters from the in situ heating steps and corresponding cooling to room temperature (RT).

Temperature (°C)	2*θ*	d-Spacing (Å)	2*θ* (RT)	d-Spacing (Å) RT	Δd-Spacing	Crystallite Size RT (Å)
RT	-	-	13.80	6.41	-	53
50	13.75	6.43	13.81	6.41	0.02	54
75	13.75	6.43	13.87	6.38	0.05	59
100	13.73	6.44	13.90	6.37	0.07	76
125	13.70	6.46	13.93	6.35	0.11	86
150	13.67	6.47	13.96	6.34	0.13	102
175	13.65	6.48	13.99	6.33	0.15	116
200	13.62	6.50	14.02	6.31	0.19	130

**Table 2 polymers-15-02277-t002:** Dielectric parameters obtained for Parylene C thin films with different thicknesses for a 2.25 cm^2^ device area.

Thickness (μm)	Capacitance per Unit Area at 100 kHz (nF/cm^2^)	Dielectric Constant at 100 kHz	tan δ at 100 kHz	Maximum E Field Tested (MV/cm)
0.24	9.48	2.60	0.028	4.11
0.46	6.21	3.24	0.030	2.16
0.96	3.00	3.25	0.027	1.04
1.14	2.48	3.19	0.035	0.88
1.45	1.83	3.20	0.024	0.69

**Table 3 polymers-15-02277-t003:** Parameters associated with X-ray diffractograms for as-deposited and 160 °C-cured single- and double-layered Parylene C.

	Peak Height (a.u.)	Peak Position (2*θ*)	FWHM (2*θ*)	d-Spacing (Å)	Crystallite Size (Å)
Single layer	252.15	13.84°	1.54	6.39	43
Double layer	164.27	13.85°	1.70	6.39	40
Single layer and cure	1691.37	14.03°	0.66	6.31	116
Double layer and cure	1277.65	14.01°	0.67	6.32	110

**Table 4 polymers-15-02277-t004:** Parameters associated with X-ray diffractograms for 160 °C-cured single- and double-layered Parylene C, before and after applying the temperature stimulus (62 °C for 2 h).

	Peak Height (a.u.)	Peak Position (2*θ*)	FWHM (2*θ*)	d-Spacing (Å)	Crystallite Size (Å)
Single layer and cure	1691.37	14.03°	0.66	6.31	116
Double layer and cure	1277.65	14.01°	0.67	6.32	110
Single layer, cure, and 62 °C for 2 h	1700.28	14.03°	0.65	6.31	116
Double layer, cure and 62 °C for 2 h	1259.11	14.00°	0.68	6.32	112

## Data Availability

The data presented in this study are available on request from the corresponding author.

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
