# Peer review of "Parylene C as a Multipurpose Material for Electronics and Microfluidics"

_polymers, 2023, doi:10.3390/polym15102277_

Round 1

Reviewer 1 Report

This paper presents a comprehensive study regarding the properties and performances of Parylene C-based dielectric layers and devices. Since Parylene C is a widely used dielectric polymer, this paper can provide various base data for practical applications. A few comments as below.

1) Figure 3 is very confusing, the different trend of change in d-spacing of heating and cooling round can lead to misunderstanding of their materials. the author should clearly explain the reason for this diffference.

2) It seems a bit strange to me that the breakdown strength of their Parylene C is so low as presented in Table 2. Can the author compare their performance with other works and explain the differences?

3) I suggest the author include the comprehensive comparisons of the performances of their devices with the state-of-art ones, which can help the reader know the potential of using Parylene C.

Reviewer 2 Report

In this paper, the authors documents and discusses research work associated with the thermal, structural, and electrical properties of Parylene C, and presenting electronic devices featuring Parylene C as a dielectric, namely thin-film transistors, metal-insulator-metal structures and digital microfluidic platforms, and further characterize them. However, certain concerns need to be addressed:

1. The paper should add some figures appropriately.

2. What the influence of the liquid on the dielectric layer in the aspect of electrowetting applications?

3. Figure 2 and figure 3 can be combined into one group figure.

4. The conclusion should condense some figures, highlighting the purpose of the paper, summarizing the methods used and the results achieved.

5. There are some grammar and word mistakes in the manuscript. Please go through the manuscript carefully again.

Reviewer 3 Report

Dear authors,

Regarding the dielectric characterization of the samples:

1. is not very clear which is the electrodes' configuration employed for performing the dielectric/electric measurements! Please provide additional info!

2. It is not very clear for me if the measurements were performed on single or multiple layer!

3. As in my opinion you did not employed a standardized configuration for the electrodes, I would not introduce statements as: "the dielectric constants are close to the expected value 307 of approximately 3 [7] and the loss tangent values are relatively low". I would rather make a qualitative comparison with some data from scientific literature!

4. Within supplimentary material, fig. 4 - the fact  that tan delta is not similar for 2 samples - I do not think it is an artefact - might be the interfacial polarization case it is about a double or multi layer sample

Please consider the above statements and address them within the paper!

Regards,

Round 2

Reviewer 2 Report

accept as it is